# Performance and Information Leakage in Splitfed Learning and Multi-Head Split Learning in Healthcare Data and Beyond

**DOI:** 10.3390/mps5040060

**Published:** 2022-07-13

**Authors:** Praveen Joshi, Chandra Thapa, Seyit Camtepe, Mohammed Hasanuzzaman, Ted Scully, Haithem Afli

**Affiliations:** 1Department of Computer Sciences, Munster Technological University, MTU, T12 P928 Cork, Ireland; mohammed.hasanuzzaman@mtu.ie (M.H.); ted.scully@mtu.ie (T.S.); haithem.afli@mtu.ie (H.A.); 2CSIRO Data61, Marsfield, NSW 2122, Australia; chandra.thapa@data61.csiro.au (C.T.); seyit.camtepe@data61.csiro.au (S.C.)

**Keywords:** distributed collaborative machine learning, split learning, multi-head split learning, parameter transmission-based distributed machine learning, privacy-preserving machine learning, information leakage in distributed learning

## Abstract

Machine learning (ML) in healthcare data analytics is attracting much attention because of the unprecedented power of ML to extract knowledge that improves the decision-making process. At the same time, laws and ethics codes drafted by countries to govern healthcare data are becoming stringent. Although healthcare practitioners are struggling with an enforced governance framework, we see the emergence of distributed learning-based frameworks disrupting traditional-ML-model development. Splitfed learning (SFL) is one of the recent developments in distributed machine learning that empowers healthcare practitioners to preserve the privacy of input data and enables them to train ML models. However, SFL has some extra communication and computation overheads at the client side due to the requirement of client-side model synchronization. For a resource-constrained client side (hospitals with limited computational powers), removing such conditions is required to gain efficiency in the learning. In this regard, this paper studies SFL without client-side model synchronization. The resulting architecture is known as multi-head split learning (MHSL). At the same time, it is important to investigate information leakage, which indicates how much information is gained by the server related to the raw data directly out of the smashed data—the output of the client-side model portion—passed to it by the client. Our empirical studies examine the Resnet-18 and Conv1-D architecture model on the ECG and HAM-10000 datasets under IID data distribution. The results find that SFL provides 1.81% and 2.36% better accuracy than MHSL on the ECG and HAM-10000 datasets, respectively (for cut-layer value set to 1). Analysis of experimentation with various client-side model portions demonstrates that it has an impact on the overall performance. With an increase in layers in the client-side model portion, SFL performance improves while MHSL performance degrades. Experiment results also demonstrate that information leakage provided by mutual information score values in SFL is more than MHSL for ECG and HAM-10000 datasets by $2\times 10^{-5}$ and $4\times 10^{-3}$, respectively.

## 1. Introduction

In recent years, the exponential development in healthcare-based sensors and our capability to handle big data has led to previously unseen growth in data collection [1]. Accumulated big data in the healthcare industry allows us to address healthcare challenges, such as precision medicine, skin-cancer detection, and stroke detection. However, the centralization of healthcare data raises privacy concerns and requires laws to regulate and safeguard them. Moreover, to avoid data misuse, several regulations, such as the General Data Protection Regulation (GDPR) [2], Personal Data Protection Act (PDP) [3], and Cybersecurity Law of the People’s Republic (CLPR) of China [4], have been introduced. Laws and ethics drafted by these governance frameworks for processing healthcare data aim to safeguard societal privacy and not hinder healthcare advances. As healthcare practitioners struggle with the enforced governance framework, they have to formulate time-consuming strategies to store the patients’ data, write ethics proposals and wait for long confirmation periods to start collaborative studies.

To accommodate such restrictions and the constraints placed by heterogeneous devices, improvised machine-learning (ML) approaches that preserve data privacy were sought. Federated learning [5] and split learning [6] are ML approaches (for training ML models) that protect the data privacy of the raw input data and offload computations at the central server by pushing a part of the computation to end devices.

Federated learning (FL) leverages distributed resources to collaboratively train an ML model (which can be a traditional ML model such as linear regression models or neuron-based deep-learning (DL)-based models). More precisely, in FL, multiple devices collaboratively offer resources to train an ML model while keeping the input data to themselves, as no input data leaves the place of its origin [7]. Recent research has shown that FL-trained models achieve comparable performance to ones trained on centralized datasets and perform better than those that only see isolated institutional data [8,9]. A successful deployment of FL promises to enable precision medicine [10,11] and solutions to healthcare challenges at a large scale [12], eventually leading to model generation, which yields unbiased decisions [13]. Federated learning still requires rigorous technical consideration to ensure the scalability and algorithmic guarantees to ensure no leakage of patients’ private data/information during or after model development. Firstly, training a large ML model in resource-constrained end devices is difficult because of limited resources [14,15]. Secondly, all participating end devices and the server should have a fully trained model. This does not preserve the model’s privacy between the server and participating clients during model training.

To overcome these drawbacks, split learning (SL) was introduced. SL enables a model to be split between the client-side and server-side portions. During training, client-side and server-side portions sequentially collaborate to develop the model [16]. Model split happens at the cut layer, a layer after which the remaining network portion goes to a different client or server. Once the training starts, the clients and the server never have access to the model updates (gradients) of each other’s model portion. This way, SL enables the training of large models in an environment with low-end devices such as internet-of-things and preserves the model’s privacy while training. In addition, it keeps the input data on its origin device (the analyst never has access to the input data). In recent years, SL has attracted much attention due to its capability to work well with low-end devices with limited computational capabilities. In addition, results obtained by split learning are comparable to models trained in a centralized setting on a healthcare dataset [17,18,19]. However, SL is only capable of dealing with one client at a time while training. This forces other clients to be idle and wait for their turn to train with the server [20].

To mitigate the drawback of FL having a lower level of model privacy between the clients and the server while training, and the inability of SL to train the ML model in parallel among participating clients, splitfed learning (SFL) has recently been proposed [20,21]. SFL combines the best of FL and SL. In this approach, an ML model is split between the client and the server (like in SL). In contrast to SL, multiple identical splits of the ML model, i.e., the client-side model portion, are shared across the clients. The server-side model portion is provided to the server. All clients perform the forward propagation in parallel and independently in each forward pass. Then, the activation vectors of the end layer (client-side model portion) are passed to the server. The server then processes the forward and back propagation for its server-side model on the activation vectors. In back propagation, the server returns the respective gradients of their activation vectors to the clients. Afterward, each client performs the back propagation on the gradients they received from the server. After each forward and backward pass, all client-side and server-side model portions aggregate their weights and form one global model, specifically, in splitfedv1. The aggregation is carried out independently on the client side (by using a fed server) and server side. In another version of the SFL called splitfedv2, the authors changed the training setting for the server-side model. Instead of aggregating the server-side model at each epoch, the server keeps training one server-side model with the activation vectors from all the clients.

Despite the improvements in SFL, model synchronization is needed at the client side, which is obtained through model aggregation and sharing. This is carried out to make one global model (a joint client-side and server-side model portion) consistent at the end of each epoch. However, model synchronization shifts the computation and communication overhead to the client-side. This would be significant if the number of clients grows significantly. In this regard, this paper studies SFL without client-side model synchronization. We call the resulting model architecture multi-head split learning (MHSL). Another aspect worth exploring is quantifying the leakage from smashed data, which flows between the client-side model portion to the server. In healthcare, any leakage in data can threaten patients’ privacy. Thereby, it becomes essential to evaluate the settings in which SFL can provide maximum privacy to patients’ confidential data. In this regard, we investigate the leakages in SFL and MHSL on the server side. Overall, we summarize our contributions under three research questions, stated in the following:

### Our Contributions

 **RQ1** Can we allow splitfed learning without client-side model synchronization?Firstly, we propose MHSL, which removes the client-side model synchronization. Then, we study the feasibility of MHSL over the healthcare IID distributed dataset ECG and HAM-10000. The result is extended to MNIST, KMNIST, and CIFAR-10 datasets for a more comprehensive study. Our empirical studies illustrate that MHSL is feasible; MHSL performance is comparable to SFL while reducing the communication and computation overhead on the client side. **RQ2** Is there any effect on the overall performance if we change the number of layers in the client-side model portions?For the selected healthcare dataset and the extended datasets, the performance of SFL and MHSL remains comparable if only the first layer forms the client-side model portion (i.e., cut-layer is set to 1) and the rest of the layers reside at the server-side model portion. However, as the number of layers is increased in the client-side model portion (i.e., cut-layer is set to 2, 3, *…*, 9), the difference in performance between the SFL and MHSL setup becomes significant. The performance of SFL improves with the increase in the number of layers in the client-side model portion, whereas MHSL performance degrades. **RQ3** How does mutual information score (measuring information leakage) behave under the setting of SFL and MHSL?We observe a relationship between information leakage and the complexity of the underlying dataset. For example, when the complexity of the dataset was lower (grayscale image classification, i.e., MNIST and KMNIST), information leakage was higher in the MHSL than in the SFL. On the other hand, with increased complexity (RGB image classification, i.e., HAM-10000 and CIFAR-10), information leakage from SFL was observed to be higher than the MHSL. For example, for HAM-10000, the MIS value of SFL observed at the epoch’s end was higher than MHSL.

## 2. Multi-Head Split Learning

For experimental purposes, we chose splitfedv2 in this paper. This makes our analysis more focused on the split-learning side of SFL. Moreover, we study if the federated learning part can be removed from SFL’s client-side, resulting in multi-head split learning (MHSL) for the distributed healthcare dataset.

The overall architecture of MHSL is depicted in Figure 1. In a simple setup, the full model W is split into two portions: the client-side model $W^{C}$ portion and the serverside model $W^{S}$ portion. For the clients, their models are represented by $W_{k}^{C}$, where k∈{1,2,⋯,N} is the client’s label. The global model W is formed by concatenating the $W^{C}$ and $W^{S}$, i.e., $\left[W^{C}W^{S}\right]$ once the training completes. The details of MHSL are presented in Algorithm 1.
**Algorithm 1:** Multi-head split learning (MHSL)
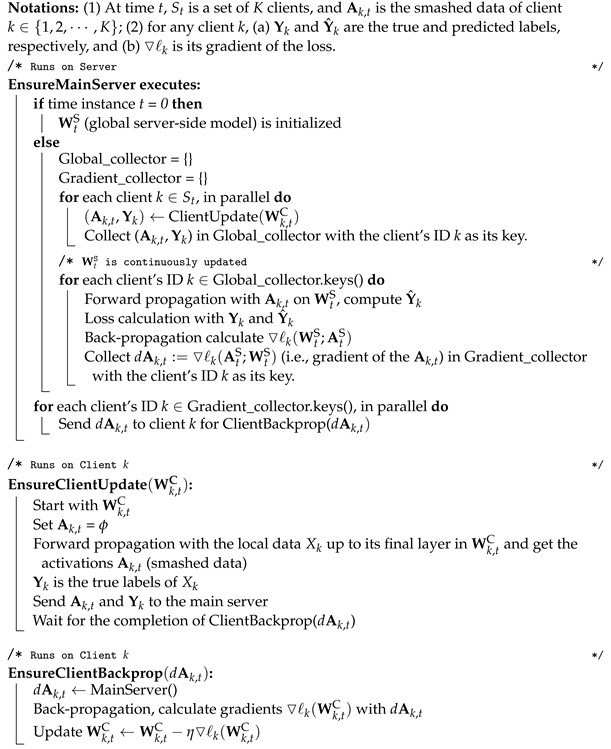


### How Is the Final Full Model Formed in Multi-Head Split Learning?

Unlike SFL, MHSL removes the fed server and the synchronization of $W_{k}^{C}$ at the end of each epoch. During the whole training, $W_{k}^{C}$ are trained independently by their clients with the server. However, at the end of the full training, the full global model W is constructed from any one $W_{k}^{C}$ and concatenated with $W^{S}$. To enable this way of constructing the final trained model, the same test data is used to evaluate each client and only keep the training data localized. Thus, if all clients’ test results are similar, it is reasonable to pick any $W_{k}^{C}$ for the final full model.

## 3. Datasets and Models

We consider image and non-image datasets and two model architectures in our studies. These are detailed in the following sections.

### 3.1. Datasets

For our research, we selected two tasks in healthcare: skin-cancer detection (image-classification problem) and ECG signal classification (time-series-based classification problem). The skin-cancer classification dataset used in this study is HAM-10000 [22]. HAM-10000 is an image-based dataset consisting of 9013 images in the training dataset and 1002 images in the test dataset. The dimension of each image in the HAM-10000 dataset is 810,000 (600×450). On the other hand, the ECG signal dataset [23] used for experimentation is a time-series-based dataset that consists of 13,245 instances both in the training and testing dataset.

In addition, we selected three widely used image datasets, MNIST, KMNIST, and CIFAR-10, for our extended experiments. Moreover, these datasets maintain our results’ closeness (as the splitfed paper uses the same datasets) with the reported results in the original paper detailing splitfedv2. The MNIST [24] dataset consists of 60,000 images in the training dataset and 10,000 images in the test dataset. The dimension of each image in the MNIST dataset is 784 (28×28) in grayscale. KMNIST [25] is another dataset used in this study that is adapted from the Kuzushiji Dataset. KMNIST consists of 60,000 images in the training dataset and 10,000 in the test dataset. The dimension of each of the images in the KMNIST dataset is 784 (28×28) in grayscale. Another dataset used for experimentation is CIFAR-10 [26], consisting of 50,000 images in the training set and 10,000 images in the test dataset. Each image corresponds to the dimension of 3072 (32×32). For the summary, refer to Table 1. All of the datasets have ten classes for prediction.

For the experimentation, horizontal flipping, random rotation, normalization, and cropping on HAM-10000, MNIST, KMNIST, and CIFAR-10 were conducted to avoid the problem of over-fitting. Whereas, for ECG, no transformations were applied. In addition, for all our experiments, data was assumed to be uniformly, independently, and identically distributed (IID) amongst five clients.

### 3.2. Models

The Resnet-18 [27] network architecture was used for the primary experimentation on the HAM-10000, MNIST, KMNIST, and CIFAR-10 datasets. The Resnet-18 network was selected because of the discrete “blocks” structure in every layer of the architecture [27], and it is a standard model for image processing. Resnet-18 blocks were used to split Resnet-18 between the clients and server to form the client-side and server-side models. Each block performs an operation; an operation in a block refers to passing an image through a convolution, batch normalization, and a ReLU activation. Resnet-18 in the experiment is initialized with a learning rate of $1\times 10^{-3}$ and the mini-batch size of B.N. was set to 256 based on the initial experimentation, Section 6.1. In addition, the first convolutional-layer kernel size was set to 7 × 7, remaining convolutional layers used 3 × 3 kernels, as shown in the model architecture in Table 2.

The Conv1-D architecture [28] was used for the primary experimentation of the ECG time-series dataset. Regarding architecture selection for ECG, the Conv1-D architecture was selected because of its efficiency in dealing with sequential data [29,30]. Another determining factor for selecting the Conv1-D architecture instead of sequential models (such as LSTM, GRU, and RNN) is that there is no effective approach for splitting the sequence model between the client and server in the SFL setting. The Conv1-D architecture comprises the discrete “blocks”. Conv1-D blocks were used to split the Conv1-D architecture between the clients and server to form the client-side and server-side models. Each block performs an operation; an operation in a block refers to passing an image through convolution and a ReLU activation. The Conv1-D architecture, in the experiment, was initialized with a learning rate of $1\times 10^{-3}$, and the mini-batch size of B.N. was set to 32 based on the initial experimentation (see Section 6.1). In addition, the kernel size of the first convolutional layer was set to 7 × 7, while a kernel size of 3 × 3 was used for all remaining convolutional layers (full architecture detail in Table 2).

Our program was written using Python 3.7.6 and PyTorch 1.2.0 libraries. The experiments are conducted in a Tesla P100-PCI-E-16GB GPU machine system. We observe the training and testing loss and accuracy at each global epoch (once the server trains with all the activation vectors received from all clients). We considered the client-level performance. All the clients were selected to participate at least once at a global epoch without repetition for the current setup.

## 4. Total Cost Analysis

Assuming *K* to be the number of clients participating in training, *p* represents the accumulated size of the data, and |A| is the size of the activations passing through the smashed layer from the client side to the server side for a single client considering one input sample. Similarly, *R* is the communication rate of the data transfer from the client to the server, *T* is the computation time incurred in one forward propagation (accumulative time taken for client-side and server-side forward propagation), and backward propagation (accumulative time taken for client-side and server-side backward propagation) with a dataset of size *p*. In addition, $T_{\mathrm{fedavg}\;}$ is the time required by fed server to perform the client-side model aggregation for all participating clients *K* (for any architecture). Let |W| be the size of the full model distributed between the client side and server side, and β defines the fraction of the full model’s size available in a client in SFL, i.e., $\left|W^{C}\right|=\beta \left|W\right|$. One forward and backward propagation requires a client to upload and download the client-side model to and from the fed server to aggregate the client-side model weights; hence, communication size becomes 2β|W| per client.

The above assumptions were used to define the communication size per client, total communication size, and overall training time for SFL and MHSL. All the related costs are presented in Table 3.

As seen in Table 3, in contrast to SFL, MHSL avoids the extra communication cost in terms of data traversal (communication size) through the network by 2βK|W|, as no client-side model aggregation is required. It also saves the computational time needed by fed server to aggregate the client-side weights by $T_{\mathrm{fedavg}}$. For our analysis of computation time, we performed computation-time measurements in our experiments. Figure 2 illustrates the average computation time logged by all the client-side and server-side model portions during one epoch. The client-side and server-side time required for one forward and backward propagation remained computationally alike when SFL was compared with MHSL. The only difference arose with additional computation requirements by fed server in order to complete the client-side model aggregation in SFL. In Figure 2, the computation time (in seconds) required by fed server (to carry out aggregation of client-side model portion) is represented as the exponential of $T_{\mathrm{fedavg}\;}$. It was carried out to scale up the values $T_{\mathrm{fedavg}\;}$, so that it can be visible in the chart. A more considerable computation time over the client-side and server-side model portion for HAM-10000 can be noticed compared to the other datasets because each image is of significantly high dimension. In addition, in MHSL, one can observe that there is no value for computation time required by fed server. *No value* for fed-server computation time is because of the MHSL setup in which we removed the client-side model-portions weight’s aggregation by fed server as compared to SFL setup. In addition, the computation time of SFL is comparable in all the datasets because fed server is only responsible for the aggregation of the client-side model portion, as at no point did we change the architecture (Resnet-18) and the number of clients (being 5) participating in SFL. This is the expected behavior of the fed server.

## 5. Threat Model

We considered the privacy of raw input data (e.g., training data) in the ML training. In other words, we aim to stop the server, which is assumed to be an *honest-but-curious* entity, inferring knowledge about the raw data from the intermediate latent vectors, i.e., smashed data, passed to it in SFL and MHSL. Moreover, the honest-but-curious server performs all operations as intended, but it only tries to infer more information about the raw data from the smashed data it receives. The server does not run any reconstruction attacks on the clients in order to generate back the raw data; instead, it can utilize some function that can reveal more information about the raw data from the smashed data. In our threat model, all the participating clients are assumed to be honest entities.

As the similarity of the smashed data to the raw input data reveals more of its information to the server, the best ML approach aims to keep the closeness between the smashed data and the raw data as far apart as possible. In other words, the information gained at the server, given the smashed data, is kept as low as possible. We call such information gain information leakage at the server side. We measured the leakage by leveraging the information-theoretic metric, which is presented in the following.

### Mutual Information Score

The mutual information score (represented as *I*, in further equations) measures how much information a random variable *X* (e.g., smashed data in our case) can reveal about another random variable *Y* (e.g., raw data in our case). For *X* and *Y* with joint distribution of p(x,y), it is defined as follows: $$I\left(X,Y\right)=\sum_{x\in X,y\in Y}p\left(x,y\right)\log\frac{p(x,y)}{p(x)p(y)}=H\left(Y\right)-H\left(Y|X\right).$$

Some important properties of the mutual information score are the following:I(X,Y)≤0 if *X* and *Y* are independent;I(X,Y)≥0 if *X* and *Y* are dependent;I(X,Y)=I(Y,X);It is invariant to invertible re-parametrization, which means that for two invertible functions ϕ and ω, I(X,Y)=I(ϕ(X),ω(Y)).

In neural network layers, considering the data processing inequality and processes, we have the following:$$I\left(X,Y_{1}\right)\geq I\left(Y_{1},Y_{2}\right)\geq I\left(X_{i},Y_{i}\right)\geq I\left(X,Y\right),$$
where *X* represents the input image and $Y_{i}$ is the output of the layer *i*, for i∈{1,2⋯,N}, and *Y* is the output of the final layer of the model. A small I(X,Y) is better from the information-leakage perspective. Unlike other metrics such as correlation, it can capture linear and non-linear associations between the random variables.

The following equation estimates the mutual information score (MIS):(1)$$I\left(X,Y\right)=\sum_{i=1}^{\left|X\right|}\sum_{j=1}^{\left|Y\right|}\frac{\left|X_{i}\cap Y_{j}\right|}{N}\log\frac{N\left|X_{i}\cap Y_{j}\right|}{\left|X_{i}\right|\left|Y_{j}\right|},$$
where |·| indicates the number of unique samples in that random variable, and *N* is the total samples.

We used the scikit-learn package and the mutual information score metric to evaluate MIS for calculating MIS against each dataset [31]. To compute MIS, only one channel was considered for the image-based dataset (i.e., HAM-10000, MNIST, KMNIST, and CIFAR-10). Furthermore, as MIS is only calculated on similar length vectors, to make the length of the input image the same as that of the smashed data, rescaling transformation on the smashed data before computing the MIS was performed. In the case of the ECG dataset, the smashed data were zero-padded.

## 6. Results

The results are divided into four parts. First, Section 6.1 presents the results obtained while training the centralized version of the Resnet-18 on the HAM-10000, MNIST, KMSNIT, and CIFAR-10 datasets and the results of a Conv1-D architecture on the ECG dataset. Secondly, in Section 6.2, we compare the results of splitfedv2 and MHSL on the HAM-10000, ECG, MNIST, KMSNIT, and CIFAR-10 datasets. We consider five clients participating in model training in order to remain consistent with the approach adopted in splitfedv2 [20]. In both architectures (Resnet-18 and Conv1-D), we kept the initial layer inside the clients (as a client-side model portion) and the rest of the layers residing in the server (as a server-side model portion). Thirdly, in Section 6.3, we present our empirical results indicating the impact of selecting different split positions in each model on the overall performance of the Resnet-18 and Conv1-D architecture. Finally, in Section 6.4, we analyze the behavior exhibited in terms of mutual information score by the SFL and MHSL setups. All the experiments are carried out for 50 epochs, excluding the one carried out on HAM-10000 dataset. The training of Resnet-18 on HAM-10000 was carried out for 15 epochs due to limited computational power.

### 6.1. Baseline Result

For the baseline, HAM-10000, MNIST, KMSNIT, and CIFAR-10 were run on the Resnet-18 architecture, whereas ECG was run on the Conv1-D architecture. Data-augmentation techniques were the same as discussed in Section 3.1, for all the datasets. Training of the Resnet-18 and Conv1-D architecture was performed in a centralized manner, i.e., the whole model resided in the server without any split, and all data were available to the server. The train and test accuracies for all datasets and corresponding models are summarised in Table 4.

### 6.2. Experiment 1: Corresponding to **RQ1**


This section evaluates SFL and MHSL. In this regard, the model was split at the first layer. The first layer resides on the client side (client-side model portion) and the remaining on the server-side (server-side model portion). Experimental results in terms of test accuracy on ECG, HAM-10000, MNIST, KMSNIT, and CIFAR-10 datasets with and without client-side aggregation are provided in Table 5.

Our empirical results, as shown in Table 5, demonstrate that the healthcare-based distributed dataset has not suffered a significant drop in accuracy. The difference in the performance of SFL over MHSL for the ECG dataset is 1.81%, whereas, for HAM-10000, it increases to 2.36%. For the healthcare dataset and CIFAR-10 datasets, we see that SFL outperforms MHSL by over 2%. In contrast, MNIST with an MHSL setup performed well and improved by 0.49% over the counterpart SFL setup. For KMNIST, although the SFL setup obtained higher accuracy, the difference between the SFL and MHSL was merely 0.06%. For CIFAR-10, the performance between SFL and MHSL was highest compared to any other dataset, which was observed to be 2.5%.

### 6.3. Experiment 2: Corresponding to **RQ2**

This section evaluates the impact of the model’s portion size on the client side on the overall performance. Test accuracy on ECG, HAM-10000, MNIST, KMNIST, and CIFAR-10 datasets is shown in Table 6.

From Table 6, it is evident that SFL and MHSL show a comparable test performance at cut-layer 1. MNIST is the dataset where MHSL performed better than SFL, but only when the cut-layer value was 1 or 2. In addition, a significant difference is seen in the performance of HAM-10000 and CIFAR-10 over cut-layer 6, 7, 8, and 9. SFL performance was better than MHSL by a difference of approximately 10–15% for CIFAR-10 and 4–8% for HAM-10000. Other datasets also exhibit the same behavior, excluding ECG, where, when a higher number of layers were introduced over the client-side, the model portion improved the performance of SFL and, at the same time, MHSL performance deteriorated. This observation helps us conclude that having more layers inside the client side significantly improves the performance of SFL but simultaneously deteriorates the MHSL performance. Overall, for IID datasets among the clients, our empirical results (both under **RQ1** and **RQ2**) demonstrate that multi-head split learning (MHSL) is feasible (it can be used to develop a production-level ML model). However, as more layers are shifted towards the client-side model part, the difference in performance becomes significant.

### 6.4. Experiment 3: Corresponding to **RQ3**

This section analyzes the MIS of the smashed data at the cut layer on the ECG, HAM-10000, MNIST, KMNIST, and CIFAR-10 datasets.

For the ECG dataset, it was observed that the MIS score oscillated within epochs. However, after the full training process, the MIS value for SFL was found to be more than the MHSL value (see Figure 3). In addition, a non-positive value of MIS was observed during the experiment, which indicates that the range of values of smashed data and input data were significantly different, which could be a case of naively padding 0 to the smashed data.

For HAM-10000 and CIFAR-10, the MIS values or information leakage was higher in SFL than MHSL, as seen in Figure 4. In contrast, MIS values for the MNIST and KMNIST datasets showed more leakage through the cut layer in MHSL when compared to SFL, which can be seen in Figure 5. Furthermore, the mutual information score (MIS) seems to exhibit a relationship with the dataset complexity. For example, for low-complexity datasets (grayscale image classification, i.e., MNIST and KMNIST), the MIS was higher for MHSL when compared with SFL. On the other hand, with increased complexity (RGB image classification, i.e., HAM-10000 and CIFAR-10), the MIS score of SFL was higher than MHSL.

The visual output of smashed data from SFL and MHSL at the cut layer is provided in Table 7 for visual inspection, along with the corresponding input images. Smashed data plotted in Table 7 is captured at the last epoch of the training phase. In addition, Resnet-18 architecture with a cut-layer value set to 3 was selected for the visualization task. One can observe that the image formed from the cut-layer’s smashed data is more visually distorted in MHSL than in SFL.

## 7. Conclusions and Future Works

This paper studied performance and information leakage in the distributed learning of healthcare data and other commonly used datasets with splitfed learning (SFL) and multi-head split learning (MHSL).

Our experiment results with Resnet-18 on HAM-10000, MNIST, KMNIST, and CIFAR-10 demonstrated that MHSL is applicable as an alternative to SFL-based ML model development for the IID distributed data. The results indicate that removing the client-side aggregation ML model can achieve comparable results to the ML model when retaining client-side aggregation. Furthermore, this approach reduces the requirement of an additional server for performing FL for client-side model aggregation. For healthcare-based datasets, ECG and HAM-10000, SFL is seen to be marginally better than MHSL (by 1.81% and 2.36%), which remained the case for extended datasets, with MNIST being the only exception, where MHSL showed a marginal improvement (of 0.49%) over SFL.

Our results demonstrated that adopting different layers to act as a split point for the network has a significant effect on overall performance. Empirical configurations with a higher number of layers located at the client-side model resulted in improved performance for SFL. In contrast, increasing the number of client-side layers led to a deterioration in the performance of MHSL. In addition, our experiments also favor the MHSL over SFL for three-channel datasets, where information leakage was found to be less in MHSL than SFL while the performance remained comparable.

This paper is the initial step in investigating the feasibility of MHSL in terms of the effect of split-network portion sizes on the overall performance and low information leakage from a cut layer in the distributed learning of various datasets, including healthcare datasets. It will be interesting to see more exhaustive experiments and theoretical analysis on the convergence guarantee with the different models, various datasets, and under a more significant number of clients in the experimental setup. In addition, experimenting with the setup for non-IID data distribution is another research direction that can be explored.

## Figures and Tables

**Figure 1 mps-05-00060-f001:**
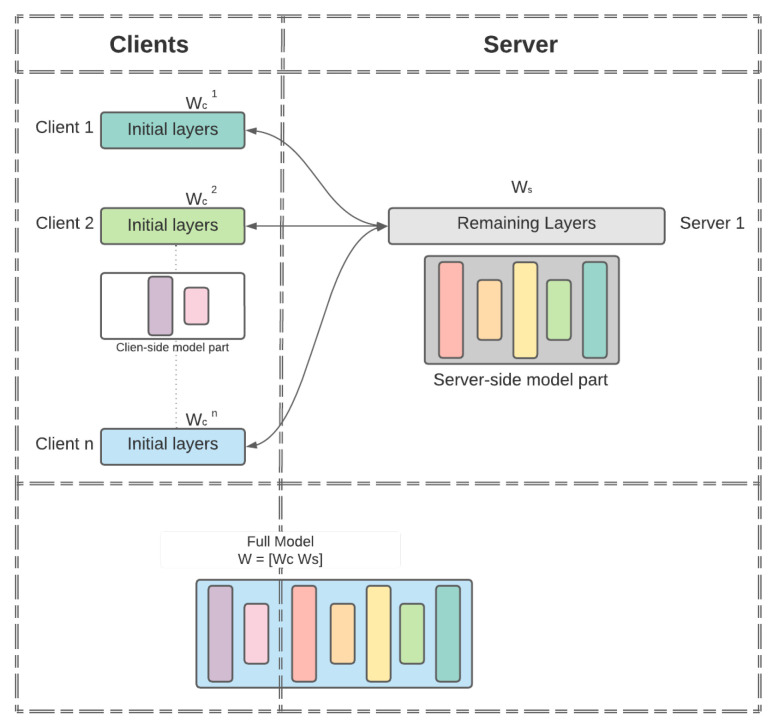
Multi-head-split-learning architecture.

**Figure 2 mps-05-00060-f002:**
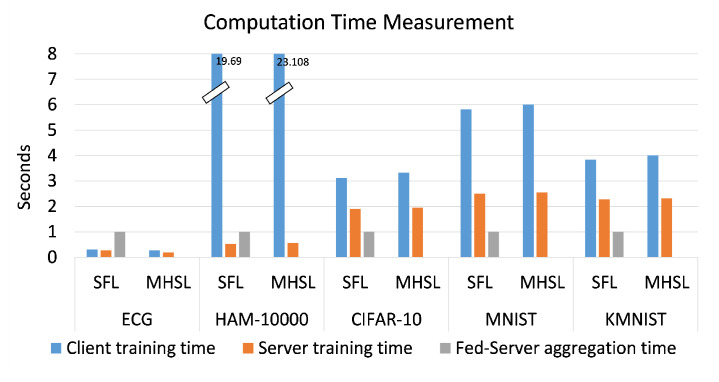
Computation time (in seconds) for SFL and MHSL.

**Figure 3 mps-05-00060-f003:**
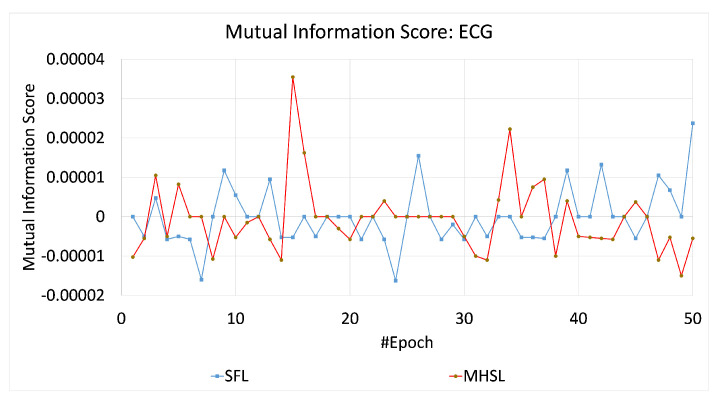
Mutual information score across the epochs for SFL and MHSL for the ECG dataset.

**Figure 4 mps-05-00060-f004:**
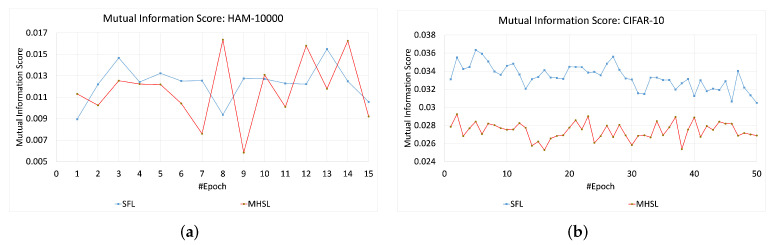
Mutual information score across the epochs for SFL and MHSL for three-channel datasets (**a**) HAM-10000 and (**b**) CIFAR-10.

**Figure 5 mps-05-00060-f005:**
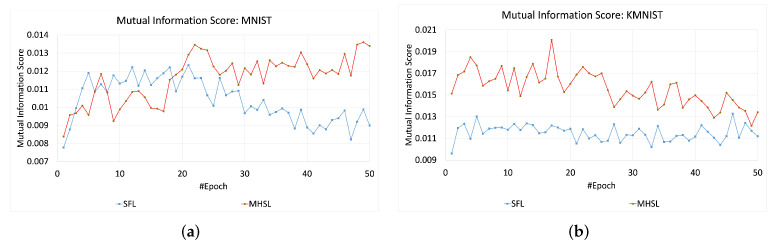
Mutual information score across the epochs for SFL and MHSL for one-channel datasets (**a**) MNIST and (**b**) KMNIST.

**Table 1 mps-05-00060-t001:** Datasets used in our experiment setup.

Dataset	Training Samples	Testing Samples	Image Size	Dataset Type	Number of Labels
ECG	13,245	13,245	NA	Time series Dataset	5
HAM-10000	9013	1002	600×450	Image Dataset	7
MNIST	60,000	10,000	28×28	Image Dataset	10
KMNIST	60,000	10,000	28×28	Image Dataset	10
CIFAR-10	50,000	10,000	32×32	Image Dataset	10

**Table 2 mps-05-00060-t002:** Model architecture used in the experimental setup.

Architecture	No. of Parameters	Layers	Kernel Size
Resnet-18 [27]	11.7 million	18	(7×7),(3×3)
Conv1-D architecture [28]	55,989	8	(7×7),(5×5)

**Table 3 mps-05-00060-t003:** Communication size and model-training-time equations for SFL and MHSL for one global epoch.

Method	Comms. Size per Client	Total Comms. Size	Total Model Training Time
SFL	$\left(\frac{2p}{K}\right)\left|A\right|+2\beta |W|$	2p|A|+2βK|W|	$T+2\frac{p|A|}{KR}+2\frac{\beta |W|}{R}+T_{\mathrm{fedavg}}$
MHSL	$\left(\frac{2p}{K}\right)\left|A\right|$	2p|A|	$T+2\frac{p|A|}{KR}$

**Table 4 mps-05-00060-t004:** Training and testing accuracy for centralized architecture.

Dataset	Model	Testing Accuracy	Training Accuracy
ECG	Conv1-D architecture	83.56	81.72
HAM-10000	Resnet-18	74.67	80.26
MNIST	Resnet-18	99.15	99.31
KMNIST	Resnet-18	95.74	99.31
CIFAR-10	Resnet-18	78.02	97.52

**Table 5 mps-05-00060-t005:** Training and testing accuracy for centralized, SFL and MHSL architectures.

Dataset	Testing Accuracy				Training Accuracy		
	Centralized	SFL	MHSL	δ (MHSL-SFl)	Centralized	SFL	MHSL
ECG	83.56	81.37	79.56	−1.81	81.72	80.98	79.43
HAM-10000	74.67	73.40	71.04	−2.36	80.26	78.80	77.77
MNIST	99.15	98.50	98.99	0.49	99.31	98.99	99.11
KMNIST	95.74	96.23	96.17	−0.06	99.31	98.98	98.99
CIFAR-10	78.02	76.25	73.75	−2.5	97.52	97.10	96.95

**Table 6 mps-05-00060-t006:** Test accuracy with respect to the model split at different layers.

Dataset	Architecture ↓ Split at Layer →	L1	L2	L3	L4	L5	L6	L7	L8	L9
ECG	**SFL**	81.37	73.45	76.57	83.90	-	-	-	-	-
ECG	**MHSL**	79.56	83.78	86.44	80.03	-	-	-	-	-
HAM-10000	**SFL**	73.40	77.25	76.59	75.46	76.45	75.43	76.35	70.89	76.69
HAM-10000	**MHSL**	71.04	71.63	73.26	71.17	72.18	72.53	71.63	71.73	68.47
MNIST	**SFL**	98.50	98.94	99.04	99.06	99.11	99.31	99.28	99.20	99.21
MNIST	**MHSL**	98.99	98.97	98.92	98.43	98.50	98.37	98.19	98.15	98.18
KMNIST	**SFL**	96.23	96.56	96.17	96.60	96.57	96.83	96.36	96.61	97.13
KMNIST	**MHSL**	96.17	96.07	95.60	95.11	93.89	92.56	92.87	92.77	91.86
CIFAR-10	**SFL**	76.25	76.10	75.73	76.76	76.72	77.60	78.06	79.04	78.82
CIFAR-10	**MHSL**	73.75	72.37	70.94	66.83	66.47	64.91	64.58	65.70	65.58

**Table 7 mps-05-00060-t007:** Visual comparison of input images against SFL and MHSL at cut-layer three during an evaluation phase.

Dataset	Input Image	SFL	MHSL
HAM-10000	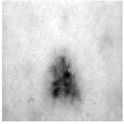	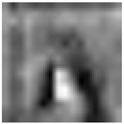	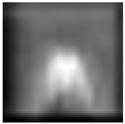
CIFAR-10	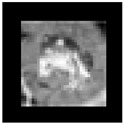	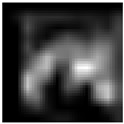	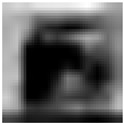
MNIST	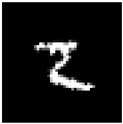	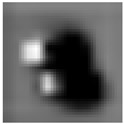	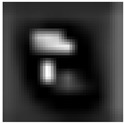
KMNIST	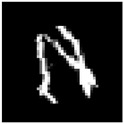	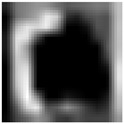	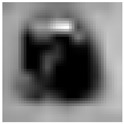

## Data Availability

This research used two healthcare datasets, and experiments were extended to the three most commonly used datasets. HAM-10000 can be found at [22]; the ECG signal dataset is provided by [23]; for the MNIST dataset, one can refer to [24]; similarly, the KMNIST dataset can be found in, [25] and the CIFAR-10 dataset can be downloaded from [26].

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
