# Peer review of "Performance and Information Leakage in Splitfed Learning and Multi-Head Split Learning in Healthcare Data and Beyond"

_mps, 2022, doi:10.3390/mps5040060_

Round 1

Reviewer 1 Report

The paper is very interesting and timely. The authors have successfully fused information leakage with DL and healthcare, which deems of increasing interest. However, some issue must be addressed as below:

  1. The title is too long, and the authors have used many confusing words together such as (distributed learning) and (splitfed learning).
  2. The authors spoke also about federated learning, but this has caused even more confusion as to what is the difference between FL and DL? You need to clarify this please.
  3. The authors need to pay more attention to the computation time given that the dataset is time-series. There are so many articles in the literature about this computation time: https://ieeexplore.ieee.org/abstract/document/8878074
  4. Figure 2, 3 and 4 look vague to hat the x-axis and y-axis refer to? This must have a caption.
  5. I feel the literature review is poor and needs more work to be added to support this paper. Although the healthcare is a key point in this paper, however, there are only 2 references used in the paper related to healthcare. Please add more.
  6. The threat model section 2.3.1 is very short and does not show any sort of analysis.
  7. The paper needs proofreading

Author Response

Greetings Reviewer,

Regards,

Praveen Joshi

Reviewer 2 Report

Godd paper, needs some corrections in spelling, grammar and correctness of data:

1. Use conequently UK or US English, no mix, e.g. cntralised, centralized, favour,...

2. Use a unique writing style for splitfed  / SplitFed  Splitfed

3. Why some data, representing the same, are different in table 3, 4, 5? (e. g. KMNIST in tabel 3: 95.74 in table 4: 95.72;                                                         ECG MHSL L1  79.55 tbab 5, 79,56 tab 4)

4. ECG is run on the Conv1-D architecture model

Author Response

(The authors gave the same response as above.)

Reviewer 3 Report

Strengths

The reasons to accept this paper are:

·         The interesting topic about the contribution of ML and DL in healthcare.

·         The study of performance and information leakage in Distributed Learning of Healthcare data and other commonly used datasets with SplitFed Learning (SFL) and Multi-head Split Learning (MHSL).

·         The introduction in which everything is well-presented and analysed.

·         The detailed presentation of Methodology.

·         The existence of detailed figures and tables in the results section making it easier to be understood.

Weaknesses

There are no major reasons to reject this paper. There are only some details that could be considered:

·         I am not sure if the contributions of the study need to be described in so much detail under each research question. Although it is helpful, it could be described in a briefer way as the results of the study are not usually described in the introduction section. This detailed part of the contributions could also be put to the conclusion section to provide the readers with a detailed take-home message. In this way, it is better to describe the contribution of this study in a better way without presenting the results in the introduction.

·         Grammatical errors. For example, there is wrong use of tenses in few parts of the paper.

·         Some spelling errors.

·         There could be some more space under the caption of figure 2.

Author Response

(The authors gave the same response as above.)

Round 2

Reviewer 1 Report

The authors address my comments. The highlighted parts prove the efforts they put in the revised version. I have no other comments.